# An Image Encryption Algorithm Based on Improved Hilbert Curve Scrambling and Dynamic DNA Coding

**DOI:** 10.3390/e25081178

**Published:** 2023-08-08

**Authors:** Shengtao Geng, Jiahao Li, Xuncai Zhang, Yanfeng Wang

**Affiliations:** School of Electrical and Information Engineering, Zhengzhou University of Light Industry, Zhengzhou 450002, China; gst@zzuli.edu.cn (S.G.); 332101050056@email.zzuli.edu.cn (J.L.); wangyanfeng@zzuli.edu.cn (Y.W.)

**Keywords:** image encryption, DWT, Hilbert curve, bit-level scramble, DNA coding, ciphertext feedback

## Abstract

As an effective method for image security protection, image encryption is widely used in data hiding and content protection. This paper proposes an image encryption algorithm based on an improved Hilbert curve with DNA coding. Firstly, the discrete wavelet transform (DWT) decomposes the plaintext image by three-level DWT to obtain the high-frequency and low-frequency components. Secondly, different modes of the Hilbert curve are selected to scramble the high-frequency and low-frequency components. Then, the high-frequency and low-frequency components are reconstructed separately using the inverse discrete wavelet transform (IDWT). Then, the bit matrix of the image pixels is scrambled, changing the pixel value while changing the pixel position and weakening the strong correlation between adjacent pixels to a more significant correlation. Finally, combining dynamic DNA coding and ciphertext feedback to diffuse the pixel values improves the encryption effect. The encryption algorithm performs the scrambling and diffusion in alternating transformations of space, frequency, and spatial domains, breaking the limitations of conventional scrambling. The experimental simulation results and security analysis show that the encryption algorithm can effectively resist statistical attacks and differential attacks with good security and robustness.

## 1. Introduction

With the fast development of the internet and multimedia technology, information security is gaining more and more attention. To prevent images from being stolen during transmission, researchers have proposed many methods for image protection, and image encryption is a common method for securing image transmission. Early encryption methods are mainly data encryption. With continuous research, researchers have proposed new encryption methods that address the drawbacks of early encryption algorithms, namely low operational efficiency, small key space, and poor security [1,2].

In 1963, Lorentz [3] introduced the concept of chaos theory, which is widely used in image encryption due to the sensitivity, ergodicity, and unpredictability of chaotic systems to initial states and control parameters [4,5,6,7]. Xu [8] proposed a new image encryption algorithm based on one-dimensional logistic mapping and the orthogonal Latin square, improving ciphertext image security. The advantages of one-dimensional logistic mapping are its simple structure, high computational efficiency, and lower level of difficulty in implementation. However, its disadvantages are the short period window, limited range of chaotic behavior, small generated key space, and vulnerability to attacks [9]. To address the insufficiency of low-dimensional chaotic systems in image encryption, Gao [10] combined two one-dimensional chaotic systems and proposed a new two-dimensional chaotic system, which uses the chaotic sequence generated by the two-dimensional chaotic system to displace the row and column pixels and then performs nonlinear diffusion of the pixels, which improves the randomness of the chaotic sequence and also enhances the resistance of the encryption algorithm to attacks. Arthi [11] added a state variable to the three-dimensional Lorenz chaotic system and constructed a four-dimensional hyperchaotic system containing two positive Lyapunov exponents. It has the advantage of iterating once to obtain multiple chaotic sequences, which is more efficient, and the system parameters can also make the key space larger and effectively resist brute force attacks. Using iterative chaotic sequences for image encryption improves the complexity and robustness of the encryption algorithm.

In image encryption, scrambling and diffusion techniques are the core part of the encryption algorithm [12]. Scramble changes the position of pixels and reduces the correlation between adjacent pixels, while diffusion randomly changes the pixel values, making the ciphertext image more chaotic. Researchers have proposed many image encryption algorithms based on scrambling and diffusion techniques, most of which perform scrambling followed by diffusion [13,14,15]. Although this encryption algorithm has good security, there are some problems. For example, in [16], the scrambling part uses only the Hilbert fill curve to scramble pixels, which does not entirely break the correlation between adjacent pixels, making it less effective and more vulnerable to brute force attacks. The single scrambling and diffusion operations are too simple, resulting in a less secure encryption algorithm [17]. In contrast, multiple scrambling and diffusion repetitions are time-consuming and significantly reduce the encryption efficiency [18]. To address the above shortcomings, researchers have combined scrambling and diffusion and proposed bit-level scrambling with simultaneous scrambling and diffusion to encrypt images [19,20,21,22]. The bit-level scrambling divides the pixel value into eight bits, disrupting the bit positions to achieve the simultaneous scrambling and diffusion of pixels. Xiang [23] proposed an image encryption algorithm that encrypts only the upper four bits of the image pixel value, which improves the encryption performance and reduces the encryption time by half. Li [24] proposed a bit-level scrambling method based on the binary tree, simultaneously changing the pixel position and value. Wang [25] used bit cyclic displacement in the scrambling phase, and the algorithm performs well through security and performance analysis.

The scrambling algorithm disrupts the pixels’ position and eliminates the correlation between them; thus, to better conceal the key information of the image, further diffusion of the pixel values is necessary. DNA coding has received great attention from more researchers because of its low power consumption, high density, and parallelism. DNA coding was first proposed by Clelland [26] in cryptography, and since then, cryptographic algorithms combining DNA coding with chaotic systems have emerged [27,28,29,30]. Other diffusion algorithms that have been proposed are the matrix half-tensor product [31], the Feistel-like network [32], filtered convolution [33], etc. In [29], Jithin divides the color image into three planes of RGB, converts these three planes into DNA base planes using fixed encoding rules, and performs heteroskedastic operations with these three DNA base planes using DNA matrices generated from chaotic sequences. In [30], Wang uses different encoding rules to convert multiple plaintext images into multiple DNA matrices. Nevertheless, each pixel has the same encoding rules for the same plaintext image matrix, performs operations with the chaotic sequence-generated DNA matrix, and uses different decoding rules. However, these DNA coding-based image encryption algorithms achieve pixel diffusion for their purposes; they also have a significant drawback, as the fixed DNA coding and decoding rules cannot change the bit distribution of pixels and are vulnerable to brute force attacks [34].

Image encryption methods have frequency domain encryption in addition to spatial domain encryption. In [35], Shafique uses multiple S-boxes combined with wavelet transform to encrypt images, which shortens the encryption time and solves the problem of a weak single S-box encryption. In [36], Yan used fractional-order wavelet transform to perform third-order fractional wavelet transform on plaintext images to obtain high-frequency and low-frequency components, index scrambling for each component using an index sequence generated by the chaotic sequence, and finally, diffusing the scrambled image using a cyclic shift. The resulting ciphertext image has good robustness. In [37], Qin uses dynamic wavelet decomposition and scrambling diffusion simultaneously to combine spatial-domain and frequency-domain encryption, ensuring both the security and robustness of the encryption algorithm.

This paper proposes an image encryption algorithm based on improved Hilbert curve scrambling and dynamic DNA coding by combining a 4D hyperchaotic system to summarize the above. Firstly, the hash value of the plaintext image is obtained using the SHA-384 algorithm, and the initial value of the hyperchaotic system is calculated. Secondly, the decomposition of the plaintext image is achieved using three-level DWT to obtain one low-frequency component and nine high-frequency components. These ten components are scrambled using different modes of the Hilbert curve, and the high-frequency and low-frequency components are then reconstructed using IDWT. Then, the bit matrix of the image pixels is position-scrambled to enhance the scrambling effect. Finally, the pixel values are further diffused using dynamic DNA coding and ciphertext feedback to improve the security of the encryption algorithm.

The rest of this paper is as follows: Section 2 introduces the 4D hyperchaos system, DWT, and the Hilbert curve; Section 3 presents the proposed encryption algorithm; Section 4 shows the experimental simulation results; Section 5 is an analysis of the various security of encryption algorithm; and Section 6 gives the conclusion.

## 2. Preparation

### 2.1. D Hyperchaotic System

Li introduced a nonlinear controller *W* into the chaotic system, which constitutes a 4D hyperchaotic system [38]; this is a nonlinear four-dimensional chaotic system that is reversible and discrete and can simultaneously generate four chaotic sequences with more complex behaviors and increased key space, enabling the encryption algorithm to effectively resist various attacks such as known plaintext attacks and brute force attacks, compensating for the small key space of the low-dimensional chaotic system. The expression of the 4D hyperchaotic system is shown in Equation (1).
(1)$$\left\{\begin{array}{l} \dot{x}=\delta \left(y-x\right)\\ \dot{y}=-xz+\tau x+\mu y-w\\ \dot{z}=xy-\theta z\\ \dot{w}=x+\varphi \end{array}\right.$$
where δ, θ, μ, τ, and φ are the parameters that affect the behavior of the hyperchaotic system, and φ∈[−0.7,0.7]. When δ=36, θ=3, μ=28, τ=16, and φ=0.2, the Lyapunov exponents $\lambda_{1}=1.552$, $\lambda_{2}=0.023$, $\lambda_{3}=0$, and $\lambda_{4}=-12.573$ of the hyperchaotic system, which contains two positive Lyapunov exponents, has better chaotic behavior. The computation time is somewhat shorter than the usual chaotic system.

Based on the above parameters, Figure 1 shows the phase diagrams of the hyperchaotic system in two and three dimensions after discretization by the 4th-order Runge–Kutta method. The phase diagram in each dimension indicates that the system has multiple attractors and is a hyperchaotic system with complex variations. In this paper, the parameters δ=36, θ=3, μ=28, and τ=16 are taken to iterate the equations of the hyperchaotic system to obtain four sequences for image encryption.

### 2.2. Discrete Wavelet Transform

The DWT is a discretization of the scales and translations of the fundamental wavelet that can decompose the signal at different scales and decompose the signal into components of different frequencies. 2D-DWT is defined by Equation (2).
(2)$$\left\{\begin{array}{c} T_{\Phi }\left(j_{0},m,n\right)=\frac{1}{M\times N}\sum_{x=0}^{M-1}\sum_{y=0}^{N-1}g\left(x,y\right)\Phi_{j_{0},m,n}\left(x,y\right)\\ T_{\Psi }^{i}\left(j,m,n\right)=\frac{1}{M\times N}\sum_{x=0}^{M-1}\sum_{y=0}^{N-1}g\left(x,y\right)\Psi_{j,m,n}^{i}\left(x,y\right) \end{array}\right.$$
where $T_{\Phi }\left(j_{0},m,n\right)$ denotes the approximate part of the image, $T_{\Psi }^{i}\left(j,m,n\right)$ denotes the edge section of the image, $\Phi_{j_{0},m,n}\left(x,y\right)$ denotes the scaling function, and $\Psi_{j,m,n}^{i}\left(x,y\right)$ denotes the wavelet function. The 2D-IDWT is defined by Equation (3).
(3)$$g\left(x,y\right)=(\begin{array}{l} \frac{1}{M\times N}\sum_{m}\sum_{n}T_{\Phi }\left(j_{0},m,n\right)\Phi_{j_{0},m,n}\left(x,y\right)\\ +\frac{1}{M\times N}\sum_{i=H,V,D}\sum_{j=j_{0}}^{\infty }\sum_{m}\sum_{n}T_{\Phi }\left(j_{0},m,n\right)\Psi_{j,m,n}^{i}\left(x,y\right) \end{array})$$

The advantage of DWT is that it eliminates the connection between pixels and is less distorted than the conventional discrete cosine transform (DCT) after multilevel wavelet decomposition. Figure 2 shows the DWT and IDWT process of the plaintext image. DWT decomposes the image to obtain the four components LL1, HL1, LH1, and HH1. IDWT is the column and row reconstruction of the resulting four components to obtain the original image [39]. Figure 3 shows the one-level, two-level, and three-level DWT. In the two-level DWT, the component LL1 continues to be decomposed into four components: LL2, HL2, LH2, and HH2. Similarly, in the three-level DWT, the component LL2 continues to be decomposed into four components: LL3, HL3, LH3, and HH3.

### 2.3. Hilbert Curve

The Hilbert curve is one of the classical space-filling curves [40], and similar space-filling curves include the Z-curve [41], Gray codes [42], etc. A Hilbert curve can linearly traverse every pixel point in the two-dimensional plane and travels each pixel point only once, according to the properties of their own spatially filled curve. The 2D Hilbert curve is a square divided equally into four little squares. The first iteration is completed by starting from the center of the bottom left square up to the center of the top right square, then right to the center of the entire right square, and then down to the center of the bottom right square in turn. The image is divided into several 2 × 2 submatrices according to the size of the image, and each sub-matrix is traversed by a first-order Hilbert curve with different directions. Each sub-matrix is traversed by a One-order Hilbert curve in a different direction, and then the first and last pixel points of each sub-matrix are connected. The Hilbert curve can traverse the whole image. Figure 4 illustrates the different orders of the Hilbert curve. In this paper, eight different filling modes are designed, based on the four starting positions of the Hilbert curve and two directions: horizontal and vertical. Taking the Two-order Hilbert curve as an example, Figure 5 shows these eight modes.

## 3. Encryption Algorithm

### 3.1. Secret Key Generation

The parameters of the chaotic system are calculated by the intermediate variable $a_{i}$ to make the encryption algorithm dependent on the key. Assuming that the size of the plaintext image is *M* × *N*, and $P_{i,j}$ is the pixel value of the plaintext, then i∈[1,N] and $j\in \left[1,M\right]$, and the parameter φ of the hyperchaotic system is calculated by Equations (4) and (5).
(4)$$\left\{\begin{array}{l} a_{1}=flood\left(\sum_{i=1,j=1}^{i=\frac{M}{2},j=\frac{N}{2}}mod\left(P_{i,j},256\right)\times \frac{4}{M\times N}\right)\\ a_{2}=flood\left(\sum_{i=1,j=\frac{N}{2}+1}^{i=\frac{M}{2},j=N}mod\left(P_{i,j},256\right)\times \frac{4}{M\times N}\right)\\ a_{3}=flood\left(\sum_{i=\frac{M}{2}+1,j=1}^{i=M,j=\frac{N}{2}}mod\left(P_{i,j},256\right)\times \frac{4}{M\times N}\right)\\ a_{4}=flood\left(\sum_{i=\frac{M}{2}+1,j=\frac{N}{2}+1}^{i=M,j=N}mod\left(P_{i,j},256\right)\times \frac{4}{M\times N}\right) \end{array}\right.$$
(5)$$\varphi =\frac{1}{2}\sin\left(mod\left({(a}_{1}-a_{2}{+a}_{3}{-a}_{4}\right),\frac{\pi }{2})\right)$$
where the intermediate variables $\left[a_{1},a_{2},a_{3},a_{4}\right]$ are obtained by the calculation of Equation (4), *flood* is the downward rounding function, and the *mod* is the mod function.

The plaintext image is input into the SHA-384 algorithm, which outputs a 384-bit binary, H. The H binary is divided into 48 groups of binary sequences of 8 bits each, i.e., $H_{k}=k_{1}{,\;k}_{2},k_{3},\cdots k_{48}$, and the initial values $x_{0}$, $y_{0}$, $z_{0}$, $w_{0}$ of the hyperchaotic system are calculated by Equations (6) and (7). The generated parameters φ and the initial values $x_{0}$,${\;y}_{0}$,$\;z_{0}$,$\;and\;w_{0}$ are substituted into the hyperchaotic system for 1000 + 4*M* × *N* times, and the first 1000 times are rounded off to eliminate the transient effect and to obtain the four chaotic sequences SX, SY, SZ, and SW. The four chaotic sequences are processed using Equation (8) to obtain the sequences X, Y, Z, and W that are used in the encryption algorithm.
(6)$$\left\{\begin{array}{l} x_{0}=\frac{mod\left({Q_{1}+Q_{2}+Q_{3}+Q}_{4},256\right)}{256}\\ y_{0}=\frac{mod\left({{Q_{3}+Q}_{4}+Q}_{5}+Q_{6},256\right)}{256}\\ z_{0}=\frac{mod\left({Q_{5}+Q_{6}+Q_{7}+Q}_{8},256\right)}{256}\\ w_{0}=\frac{mod\left({Q_{2}+Q_{4}+Q_{6}+Q}_{8},256\right)}{256} \end{array}\right.$$
(7)$$Q_{i}=k_{6*i-5}⨁k_{6*i-4}⨁k_{6*i-3}⨁k_{6*i-2}⨁k_{6*i-1}⨁k_{6*i}$$
where $\left[Q_{1}\sim Q_{8}\right]$ is the intermediate variable, 1≤i≤8, and ⨁ is the XOR operation.
(8)$$\left\{\begin{array}{l} X\left(i\right)=mod\left(SX\left(i\right)\times 10^{12},256\right)\\ Y\left(i\right)=mod\left(SY\left(i\right)\times 10^{12},256\right)\\ Z\left(i\right)=mod\left(SZ\left(i\right)\times 10^{12},256\right)\\ W\left(i\right)=mod\left(SW\left(i\right)\times 10^{12},256\right) \end{array}\right.1\leq i\leq 4M\times N$$

### 3.2. Pixel Scrambling

#### 3.2.1. Hilbert Curve Scrambling

To completely disrupt the image pixel location distribution, a pixel-level scrambling method is proposed. DWT is used to decompose the plaintext image into four components—${LL}_{1}$, ${HL}_{1}$, ${LH}_{1}$, and ${HH}_{1}$—and the results are shown in Equation (9).
(9)$$DWT\left(P\right)=\left[{LL}_{1},{HL}_{1},{LH}_{1},{HH}_{1}\right]$$
where *P* is the plaintext, and ${LL}_{1}$ is the low-frequency component. ${HL}_{1}$, ${LH}_{1}$, and ${HH}_{1}$ are the high-frequency components. To reduce the redundancy and encryption time, only the low-frequency components need to be scrambled with complex behavior. ${LL}_{1}$ is used as the new matrix for two-level DWT to obtain four components—${LL}_{2}$, ${HL}_{2}$, ${LH}_{2}$, and ${HH}_{2}$. Decomposition of ${LL}_{2}$ is continued for three-level DWT, and the result is shown in Equation (10).
(10)$$\left\{\begin{array}{l} DWT\left({LL}_{1}\right)=\left[{LL}_{2},{HL}_{2},{LH}_{2},{HH}_{2}\right]\\ DWT\left({LL}_{2}\right)=\left[{LL}_{3},{HL}_{3},{LH}_{3},{HH}_{3}\right] \end{array}\right.$$

The plaintext image *P* is decomposed by three-level DWT to obtain the low-frequency component ${LL}_{3}$ and nine high-frequency components, ${HL}_{1}$, ${LH}_{1}$, ${HH}_{1}$, ${HL}_{2}$, ${LH}_{2}$, ${HH}_{2}$, ${HL}_{3}$, ${LH}_{3}$, and ${HH}_{3}$. The sequence FX is obtained by intercepting the first 3M elements of the sequence X, and then the sequence FX is processed by using Equation (11). The sequence DX is divided into three sub-sequences, ${DX}_{1}$,${\;DX}_{2}$, ${DX}_{3}$, and the high-frequency components ${HL}_{1}$, ${LH}_{1}$, and ${HH}_{1}$ are scanned for scrambling with the sub-sequence ${DX}_{1}$ to select different modes of $a\left[{(\log}_{2}M)-1\right]$-order Hilbert curve. The sub-sequence ${DX}_{2}$ is used to select different modes of a $\left[{(\log}_{2}M)-2\right]$-order Hilbert curve to scan for scrambling of the high-frequency components ${HL}_{2}$, ${LH}_{2}$ and ${HH}_{2}$; then, the sub-sequence ${DX}_{3}$ is used to select different modes of a $\left[{(\log}_{2}M)-3\right]$-order Hilbert curve to scan for the scrambling of the components ${HL}_{3}$, ${LH}_{3}$, ${HH}_{3}$, and ${LL}_{3}$. The pixels on the scan path are arranged into two-dimensional matrices by rows to obtain the scrambled matrices ${Shl}_{1}$, ${Slh}_{1}$, ${Shh}_{1}$, ${Shl}_{2}$, ${Slh}_{2}$, ${Shh}_{2}$, ${Shl}_{3}$, ${Slh}_{3}$, ${Shh}_{3}$, and ${Sll}_{3}$. Finally, the ten scrambled matrices are reconstructed by Equation (12) to obtain the scrambled matrix $P^{\prime }$. Assuming that the size of the low-frequency component ${LL}_{2}$ is 8 × 8, Figure 6 shows the Hilbert curve scrambling process.
(11)$$DX=mod\left(FX,8\right)+1$$
(12)$$\left\{\begin{array}{c} {Sll}_{2}=IDWT\left({Sll}_{3},{Shl}_{3},{Slh}_{3},{Shh}_{3}\right)\\ {Sll}_{1}=IDWT\left({Sll}_{2},{Shl}_{2},{Slh}_{2},{Shh}_{2}\right)\\ P^{\prime }=IDWT\left({Sll}_{1},{Shl}_{1},{Slh}_{1},{Shh}_{1}\right) \end{array}\right.$$

#### 3.2.2. Bit-Level Scrambling

Although random pixel positions are scrambled, the pixel values are not changed, and the statistical attack can still obtain valid information in plaintext to resist statistical attacks. To resist statistical attacks and prevent an encrypted image from being cracked, each pixel value needs to be changed. To solve this problem, this paper uses the main-diagonal extraction model to scramble the position of the pixel’s bits and thus change each pixel value.

The Sarrus rule is a standard method to expand second-order and third-order determinants in linear algebra theory. In the determinant, the elements on the main-diagonal from the upper left corner to the lower right corner are called main-diagonal elements. The sub-diagonal is from the upper right corner to the lower left corner, and the elements on the sub-diagonal are called sub-diagonal elements. Since the calculation method is the product of the main-diagonal elements minus the development of the sub-diagonal elements, it is also called the diagonal rule. Taking the third-order determinant as an example, Figure 7 shows the expansion of the third-order determinant.

Based on the connection of the three favorable terms of the main-diagonal in the third-order determinant, this connection can be extended to the eighth-order to create a model of the eighth-order main-diagonal extraction, and the specific transformation process is shown in Figure 8. This matrix is partitioned into two upper and lower triangles according to the main-diagonal, and the upper triangle is extracted in the direction of the main-diagonal. In comparison, the lower triangle is removed in the opposite direction of the main-diagonal. By connecting the elements with the same color in the order of an upper triangle and then a lower triangle, eight one-dimensional sequences of length eight can be obtained, and these eight one-dimensional sequences are arranged in order in rows to form an 8 × 8 bit matrix. This model can scramble the bit matrix of image pixels.

The pixels of an image matrix *P* of size *M* × *N* are arranged in rows to form a one-dimensional sequence $P^{\prime }$, and the intercept of the last *M × N* elements of the sequence *X* is used to obtain the sequence LX. The sequence LX is then arranged in ascending order to obtain its index sequence ILX. This ILX index sequence is used to randomly selected eight pixels in the sequence $P^{\prime }$ and convert them into binary form, and each binary sequence is then arranged in order by rows to obtain the 8 × 8 bit matrix $B_{P}$. Then, the main-diagonal extraction model is used to scramble the bit matrix $B_{P}$ to obtain the bit matrix ${B^{\prime }}_{P}$. The scrambled bit matrix ${B^{\prime }}_{P}$ is converted into a decimal by rows, and this process is repeated *M* × *N*/8 times until all the bits of pixels have been scrambled to obtain the scrambled matrix $P^{\prime \prime }$. Figure 9 shows the process of the bit-level scrambling for any eight pixels.

### 3.3. Pixel Diffusion

#### 3.3.1. Dynamic DNA Coding

In biology, DNA consists of four bases: A (adenine), C (cytosine), G (guanine), and T (thymine). A and T as well as C and G are complementary. In binary, 00 and 11 and 01 and 10 are also complementary. Therefore, the four bases A, G, C, and T can be used to represent 00, 01, 10, and 11. There are 24 coding rules, of which only 8 satisfy the Watson–Crick complementarity rule [43], as Table 1 shows.

Pixel diffusion is an algorithm that changes the pixel value of an image, and it plays a crucial role in combating differential attacks. Coding a binary sequence using any DNA coding rule and decoding it using a different DNA decoding rule can change the pixel value. However, such fixed coding and decoding rules are poorly randomized, vulnerable to attacks, and lack security. Meanwhile, if fixed DNA coding rules are used for binary sequences containing consecutive ‘0’ or ‘1’ values, consecutive identical bases will occur, posing a significant security risk. To overcome this drawback and to further improve the security of ciphertext images, the binary sequences with different pixel-value bit crossovers and various DNA coding and decoding rules are randomly selected in combination with chaotic sequences, thus changing each pixel value. Dynamic DNA coding for diffusion is more random and has a better diffusion effect.

Four pixels are selected in the image matrix using Equation (13) to form a 2 × 2 sub-block, and a total of M×N/4 sub-blocks can be selected. The four pixels in a sub-block are converted into a binary sequence, and the binary sequence of pixels $P\left(i,j\right)$ and $P\left(i,N+1-j\right)$ are concatenated in the first row, followed by concatenation of the binary sequence of pixels $P\left(M+1-i,j\right)$ and $P\left(M+1-i,+1-j\right)$ in the second row. The bits of these two rows of binary sequences crossover to perform an information fusion and hide the detailed information of the pixels. The sequences Y and Z are then processed with Equation (14) to obtain the sequences DY and DZ, which are coded by the sequence DY with randomly selected DNA coding rules for bit crossover pairs, and then decoded by the sequence DZ with randomly selected DNA decoding rules. Lastly, the decoded binary sequence is converted into decimal form. The specific process is shown in Figure 10.
(13)Pi,jPi,N+1−jPM+1−i,jPM+1−i,+1−j 1≤i≤M 1≤j≤N
(14)$$\left\{\begin{array}{l} DY=mod\left(Y,8\right)+1\\ DZ=mod\left(Z,8\right)+1 \end{array}\right.$$

#### 3.3.2. Two-Way Ciphertext Diffusion

To further improve the diffusion effect, a slight change is diffused to each pixel value of the ciphertext. We use ciphertext diffusion to modify each pixel value. Sequences FW and LW are obtained by intercepting the first and last M×N values of the sequence W, respectively. The sequences FW and LW are converted into the two-dimensional matrices U and V with M rows and N columns, respectively.

Positive diffusion:(15)$$E_{i,j}=\left\{\begin{array}{cc} {mod((P}_{i,j}+P_{M,N}),\;256)⨁U_{i,j} & i=1,\;j=1\\ {mod((P}_{i,j}+E_{i,j-1}),\;256)⨁U_{i,j} & j\neq 1\\ {mod((P}_{i,j}+E_{i-1,N}),\;256)⨁U_{i,j} & i\neq 1,\;j=1 \end{array}\right.$$

Reverse diffusion:(16)$$C_{i,j}=\left\{\begin{array}{cc} {mod((E}_{i,j}+E_{1,1}),\;256)⨁V_{i,j} & i=M,\;j=N\\ {mod((E}_{i,j}+C_{i,j+1}),\;256)⨁V_{i,j} & j\neq 1\\ {mod((E}_{i,j}+C_{i+1,1}),\;256)⨁V_{i,j} & i\neq M,\;j=N \end{array}\right.$$
where i is the row, and j is the column, P is the plaintext, E represents the result of forward diffusion, and C is the ciphertext.

### 3.4. Encryption Algorithm

This encryption algorithm encrypts a square image; if the input image is non-square, it needs to be filled with “0” to make it a square with maximum side length.

The encryption algorithm proposed first obtains the key to the chaotic system using the plaintext image and the SHA-384 algorithm. A three-level DWT on the plaintext image and a Hilbert curve with different modes are randomly selected to scramble the pixels of each component locally, and then all components are reconstructed by IDWT. After that, the main-diagonal extraction model combined with the index sequence is used to scramble and diffuse each pixel simultaneously to achieve global scrambling. Finally, the scrambling matrix is modified by dynamic DNA coding and two-way ciphertext diffusion for each pixel value to obtain the ciphertext image. The encryption steps are as follows:

Input: Plaintext image P.

Output: Ciphertext image C.

**Step 1:** Input a plaintext image P of size M×N.

**Step 2:** The hash value H of the plaintext image P is calculated using the SHA-384 algorithm, and the parameters φ of the chaotic system are computed according to Equations (4) and (5). The initial values $x_{0}$, $y_{0}$, $z_{0}$, and $w_{0}$ of the chaotic system are computed according to Equations (6) and (7).

**Step 3:** Substitute the parameters and initial values into the hyperchaotic system for 4M×N+1000 iterations and remove the first 1000 values. Obtain four chaotic sequences SX, SY, SZ, and SW, which are processed according to Equation (9) to obtain the sequences X, Y, Z, and W.

**Step 4:** The three-level DWT is performed on the plaintext image P to obtain the low-frequency component ${LL}_{3}$ and nine high-frequency components: ${HL}_{1}$, ${LH}_{1}$, ${HH}_{1}$, ${HL}_{2}$, ${LH}_{2}$, ${HH}_{2}$, ${HL}_{3}$, ${LH}_{3}$, and ${HH}_{3}$. The pixels of these ten components are scrambled according to the Hilbert curve scramble in Section 3.2.1 to obtain ten scrambled matrices: ${Shl}_{1}$, ${Slh}_{1}$, ${Shh}_{1}$, ${Shl}_{2}$, ${Slh}_{2}$, ${Shh}_{2}$, ${Shl}_{3}$, ${Slh}_{3}$, ${Shh}_{3}$, and ${Sll}_{3}$. IDWT reconstructs these to obtain the scrambled matrix $P_{1}$.

**Step 5:** Arrange the pixels in the matrix $P_{1}$ by rows into a one-dimensional sequence ${SP}_{1}$. According to the bit-level scramble method in Section 3.2.2, use the index sequence ILX in the sequence ${SP}_{1}$ to randomly select eight pixels at a time for the bit-level scramble, repeating this M×N/8 times to obtain the scrambled matrix $P_{2}$.

**Step 6:** According to the dynamic DNA coding method in Section 3.3.1, four pixels in the matrix $P_{2}$ are first selected and converted into two binary sequences and allowed to undergo a bit crossover. Code the two bits of crossover with the sequence DY to obtain a DNA base sequence, which is then decoded with sequence DZ to obtain a binary sequence. This is then turned into a decimal form. Repeat M×N/4 times to obtain the diffusion matrix $P_{3}$.

**Step 7:** Diffusion methods are as described in Section 3.3.2. The matrix $P_{3}$ and the matrices U and V are globally diffused using Equations (15) and (16) to obtain the ciphertext image C.

Figure 11 shows the complete steps of the encryption algorithm.

## 4. Experimental Simulation Results

Simulation of plaintext images of Lena, Jokul, Bridge, Bank, and Peppers was performed using the proposed encryption algorithm in the MATLAB experimental platform, and the simulation results are shown in Figure 12. It is obvious that the ciphertext image loses the feature information, and the image cannot be recognized. The decrypted image can be fully recovered without distortion, which proves that the algorithm encrypts very well and has high security.

## 5. Security Analysis

The feasibility of an encryption algorithm cannot be judged by the degree of blurring of the ciphertext image alone but also by more refined experimental analyses. To test whether the proposed encryption algorithm can withstand malicious attacks by unscrupulous elements, this section compares tests in terms of key, histogram, Chi-square, correlation, information entropy, local information entropy, homogeneity, contrast, energy, PSNR, MES, and robustness.

### 5.1. Key Space Analysis

To illegally obtain some valuable information, unscrupulous people often use brute force attacks to decrypt the images being transmitted. The key space is the set of keys for the image, which is a direct criterion to judge whether an encryption algorithm has the ability to resist malicious attacks and ensure that data information is not compromised. In general, an encryption algorithm has a key space of not less than $2^{100}$. The encryption algorithm’s security becomes stronger as the key space increases. In the encryption algorithm proposed, SHA-384 is used to generate a key with a length of 384 bits, and its key space ${{KS}_{1}=2}^{192}$. The initial values $x_{0}$, $y_{0}$, $z_{0}$, and $w_{0}$ are calculated with a precision of $10^{15}$, and its keyspace ${{KS}_{2}=10}^{60}$. The total keyspace $KS={KS}_{1}\times {KS}_{2}\approx 6.28\times 10^{117}$, i.e., $KS\gg 2^{100}$, so the encryption algorithm proposed is sufficient to resist various violent attacks.

### 5.2. Key Sensitivity Analysis

A secure image encryption algorithm with ample space for the key should also have strong sensitivity. During sensitivity testing of the key, only minor changes are made to the original key, which is then used to decrypt the encrypted image. The greater the difference between the plaintext image and the ciphertext image, the stronger the sensitivity of the key.

Key sensitivity test with the Lena image: The initial parameters $x_{0}$, $y_{0}$, $z_{0}$, and $w_{0}$ are slightly changed and decrypted with the changed initial parameters. Figure 13 shows the test results, and it is apparent from Figure 13c–f that the decrypted images are entirely different, even though only minor changes were made to the key. Also, Figure 13g–l show no difference between any two decrypted images in Figure 13c–f, thus indicating that the algorithm is susceptible to the key.

### 5.3. Histogram Analysis

The distribution of the image pixels can be reflected visually from the histogram. All characteristic statistical attacks break the image by analyzing a histogram with a comparatively uneven distribution. The highly secure encryption algorithm resists statistical attacks and completely breaks up the image pixels, making the histogram more uniform. The histograms of different images are shown in Figure 14. It is not difficult to see that the distribution of ciphertext image pixels is more uniform than that of plaintext image pixels, which better hides the image information and proves that the encryption algorithm proposed can resist statistical attacks well.

### 5.4. Chi-Square Test

The intuitive histogram is only a rough assessment of the homogeneity of the image; to accurately quantify the uniformity of the histogram, a numerical operation using the difference square formula is required, which is defined by Equation (17):(17)$$s^{2}=\frac{1}{256}\sum_{0}^{255}\frac{{\left(f_{i}-g\right)}^{2}}{g}$$
where $s^{2}$ represents the Chi-square, $f_{i}$ is the appearance rate of this gray-level pixel value in the histogram of the ciphertext image, and g is the theoretical rate of appearances of the pixel value at that gray level in the histogram, denoted as g=(M×N)/256. The significance level α is chosen to be 0.05, $s_{0.05}^{2}=293.24783$. When the Chi-square of the test ciphertext image is less than this, i.e., $s^{2}<s_{0.05}^{2}$, the histogram is approximately uniformly distributed. Table 2 shows the Chi-square test results. By comparison, the ciphertext image has a much lower Chi-square value than the plaintext image, suggesting that ciphertext images have uniform pixel values.

### 5.5. Correlation Analysis

Adjacent pixels in all directions are highly correlated, and if a piece of information is compromised, statistical attacks can decipher other information based on this information, causing a chain reaction. To ensure that the image is not cracked, successfully reducing the correlation between adjacent pixels becomes the key to the encryption algorithm. The correlation coefficient is calculated using Equation (18).
(18)$$r_{x,y}=\frac{\mathrm{cov}\left(x,y\right)}{\sqrt{D\left(x\right)D\left(y\right)}}$$

The formula for each parameter in Equation (18) is as follows in Equation (19).
(19)$$\left\{\begin{array}{c} E\left(x\right)=\frac{1}{N}\sum_{i=1}^{N}x_{i}\\ D\left(x\right)=\frac{1}{N}\sum_{i=1}^{N}{\left[x_{i}-E\left(x\right)\right]}^{2}\\ \mathrm{cov}\left(x,y\right)=E\left(\left[x-E\left(x\right)\right]\left[y-E\left(y\right)\right]\right) \end{array}\right.$$
where $r_{x,r}$ is the correlation coefficient, x and y are a pair of pixel values, E(x) is the expectation of x, D(x) is the variance of x, cov(x,y) is the covariance, and N is the total number of pixels in the image. From plaintext and ciphertext images, 5000 pairs of pixels are randomly selected, and their correlation coefficients in each direction are calculated separately. The results are shown in Table 3, which shows that the correlation coefficient of the ciphertext images tends to be 0. This shows that the proposed encryption algorithm can effectively weaken the pixel correlation. Figure 15 also shows that the correlation between adjacent pixels of the ciphertext image is broken. Meanwhile, the comparison of the correlation under different encryption algorithms is shown in Table 4. The results show that the proposed encryption algorithm in this paper is more destructive to the correlation of the plaintext image compared with other encryption algorithms.

### 5.6. Information Entropy and Local Information Entropy

Information entropy is a key metric for detecting the randomness of image pixels and for quantifying the average amount of information in the image. In images with high information entropy, pixels are distributed more uniformly; the stronger the randomness, the less it is likely to be cracked. Information entropy is calculated by Equation (20):(20)$$H\left(m\right)=-\sum_{0}^{L-1}P\left(m_{i}\right)\log_{2}P\left(m_{i}\right)$$
where L is the image’s gray level. When L=256, its information entropy in the ideal state is 8. $m_{i}$ is the *i*th pixel value of the image, and $P(m_{i})$ is the chance of occurrence of the corresponding pixel value. In the information entropy test, five different images are converted into ciphertext images separately using the proposed encryption algorithm. Then, their information entropy is obtained, and the test results are displayed in Table 5. All ciphertext images have an information entropy value close to 8. Meanwhile, the comparison with other algorithms in Table 6 also visually proves that the proposed encryption algorithm has high security.

For ciphertext images with uniform pixel distribution, the calculated results are accurate, but some ciphertext images may have an uneven local pixel distribution. To overcome this drawback of information entropy and to further improve the accuracy of this evaluation criterion, Wu proposed the calculation method of local information entropy [47], as shown in Equation (21).
(21)$$H_{\left(k,T_{B}\right)\left(S\right)}=\sum_{i=1}^{k}\frac{H\left(S_{i}\right)}{k}$$
where $S_{i}$ is the ciphertext information entropy, k is the number of selected groups, and $T_{B}$ is the number of pixels in each group. The ideal range of the local information entropy is obtained when k is chosen to be 3, $T_{B}$ is 1936, and the significance level α is 0.05 in the field [7.901515698, 7.903422936]. If the test result is within this range, it means that the ciphertext image passes the test. In this test section, five different images are converted into ciphertext images using the proposed encryption algorithm; then, their local information entropy is obtained, and the test results are in Table 7. All the ciphertext images pass the test.

### 5.7. Homogeneity Analysis

The gray level co-occurrence matrix (GLCM) represents the various combinations of pixel luminance. Homogeneity analysis quantifies the distribution of elements in the GLCM and further determines how similar they are to the diagonal. The homogeneity value decreases as the distance of the elements from the diagonal becomes more prominent in the range of [0, 1], and the smaller the homogeneity value is, the more efficient the encryption algorithm is. It is calculated as shown in Equation (22) [48].
(22)$$Homogeneity=\sum_{i}\sum_{j}\frac{P\left(i,j\right)}{1+\left|i-j\right|}$$
where i and j are two horizontally adjacent gray values, and P(i,j) is the element’s value in the normalized GLCM. The homogeneity values of different plaintext images and ciphertext images are calculated, and the results are shown in Table 8. From Table 8, it can be seen that the homogeneity value of the ciphertext image is at a shallow level. Also, in Table 9, by comparing with other algorithms, the ciphertext image in this paper has the lowest homogeneity value, which shows that the encryption algorithm strongly resists statistical attacks.

### 5.8. Contrast Analysis

The contrast is usually measured for the intensity between an image pixel and its neighboring pixels. Generally, the contrast value of plaintext images is shallow, while the contrast value of ciphertext images is high. The higher the contrast value, the higher the randomness of the ciphertext image and the more resistant the encryption algorithm is to statistical attacks. The contrast value is calculated as shown in Equation (23) [48].
(23)$$Contrast=\sum_{i,j=0}P_{i,j}\times {\left(i-j\right)}^{2}$$
where i and j are two horizontally adjacent gray values, and P(i,j) is the element’s value in the normalized GLCM. The contrast values of plaintext and ciphertext images are shown in Table 10, from which it can be seen that the ciphertext image has a higher contrast value than the plaintext image. It can also be seen in the comparison in Table 9 that the ciphertext image of this paper has the highest contrast value, which shows that the encryption algorithm is effective against statistical attacks.

### 5.9. Energy Analysis

The energy is the cumulative sum of the squares of all elements in the GLCM and represents how much image information is contained. The larger the energy value of an image, the more information it contains, and the easier it is to be cracked by a statistical attack. An encryption algorithm with strong resistance to statistical attacks should have very low energy for the ciphertext image. The energy is calculated as shown in Equation (24) [48].
(24)$$Energy=\sum_{i}\sum_{j}{P\left(i,j\right)}^{2}$$
where i and j are two horizontally adjacent gray values, and P(i,j) is the element’s value in the normalized GLCM. The energy values of the plaintext image and the ciphertext image are shown in Table 11, from which it can be seen that the ciphertext image has a shallow energy value. The results of comparing the energy values of this encryption algorithm with other algorithms are shown in Table 9. It is easy to see that the energy value of the ciphertext image of this encryption algorithm is one of the lowest, which shows that this encryption algorithm has a certain degree of security in terms of resistance to statistical attacks.

### 5.10. MES and PSNR Analyses

PSNR (Peak Signal Noise Ratio) and MSE (Mean Square Error) are two objective metrics used to evaluate image quality. The more significant the PSNR value, the smaller the image distortion; the more precise the image, the worse the encryption effect. As a result, higher MSE values indicate a better encryption performance when testing plaintext and ciphertext images. They are calculated according Equations (25) and (26).
(25)$$\mathrm{MSE}=\frac{1}{M\times N}\sum_{i=1}^{M}\sum_{j=1}^{N}{\left(P\left(i,j\right)-C\left(i,j\right)\right)}^{2}$$
(26)$$\mathrm{PSNP}=10\times \log_{10}\left(\frac{L^{2}}{\mathrm{MSE}}\right)$$
where P is the plaintext, C is the ciphertext, M×N is the size of the images, and L is the pixel gray level. The MSE and PSNR values of different images after encryption and the comparison with other algorithms are shown in Table 12. From this, we can see that the PSNR of the ciphertext image is very small, and it also outperforms other algorithms when compared and analyzed against different algorithms, which indicates the high performance of this encryption algorithm.

### 5.11. Differential Attack Analysis

The differential attack mainly involves making a small change to a pixel value of the plaintext image, then encrypting the two plaintext images using an encryption algorithm, and finally comparing and analyzing the two ciphertext images to discover their connection, from which the images can be cracked.

There are two extremely critical factors in evaluating differential attacks. One is the rate of change of pixel values, NPCR, and the other is the uniform average change intensity, UACI, and these are calculated as shown in Equation (27):(27)$$\left\{\begin{array}{l} D\left(i,j\right)=\left\{\begin{array}{ll} 0 & C_{1}\left(i,j\right)=C_{2}\left(i,j\right)\\ 1 & C_{1}\left(i,j\right)\neq C_{2}\left(i,j\right) \end{array}\right.\\ NPCR=\frac{\sum_{i,j}D\left(i,j\right)}{M\times N}\times 100\%\\ UACI=\frac{1}{255\times M\times N}\sum_{i,j}\left|C_{1}\left(i,j\right)-C_{2}\left(i,j\right)\right|\times 100\% \end{array}\right.$$
where M×N is the scale size of the ciphertext, $C_{1}$ and $C_{2}$ are the two ciphertexts to be compared, and $D\left(i,j\right)$ is used to discriminate $C_{1}$ and $C_{2}$.Under ideal conditions, the values of NPCR and UACI were 99.6049% and 33.4635%, respectively. With the key unchanged, the two plaintext images are converted into ciphertext images with the encryption algorithm. Table 13 and Table 14 show the calculated NPCR and UACI values and the comparison results with different algorithms. Compared with other algorithms, this algorithm’s NPCR and UACI values are closer to the theoretical values than most of them.

### 5.12. Noise Attack Analysis

Ciphertext is vulnerable to noise attacks during data transmission. The typical noise attacks are gaussian noise and pepper noise, and the noise attacks can damage the ciphertext image and reduce the clarity. Since pepper noise has more impact on ciphertext images than other noise, in this study, different strengths of pepper noise were included for testing. While keeping the key unchanged, we added pepper noise with intensities of 0.01, 0.05, 0.1, and 0.15 to interfere with the Lena image and encrypted and decrypted it using the encryption algorithm proposed. Figure 16 shows the ciphertext and decrypted images. It is evident that even when the noise intensity reached 0.15, the decrypted image could still be recognized, indicating that the encryption algorithm resists noise attacks.

### 5.13. Cropping Attack Analysis

If the decryption algorithm is not robust against cropping attacks, the decryption will fail due to the missing information in the decryption process. If the decryption algorithm can restore the plaintext image to a large extent, the encryption algorithm is highly resistant to cropping attacks. In the test, the Lena ciphertext image was decrypted after cropping attacks of 1/64, 1/16, 1/4, and 1/2, respectively, and the results are shown in Figure 17. The Lena ciphertext image can still be seen as the basic outline of the original image after decryption with the addition of different degrees of cropping attacks, which indicates that the encryption algorithm strongly resists a cropping attack.

## 6. Conclusions

In this paper, an image encryption algorithm based on improved Hilbert curve scrambling and dynamic DNA coding is proposed. First, the image matrix is divided into ten component matrices by using three-level DWT of the plaintext image, and the chaotic sequence generated by iterating with the hyperchaotic system randomly selects one of the eight Hilbert curve modes and scrambles each of the ten component matrices. The scrambled component matrices are then reconstructed with IDWT to obtain the scrambled image matrix. Next, eight pixels at a time are randomly selected with a chaotic sequence and bit-level scrambled using the main-diagonal extraction model until all pixels are scrambled. Then, bit crossover is performed between different pixels, and the pixel values are modified using dynamic DNA coding. Finally, the pixels are globally diffused using two-way ciphertext feedback to obtain the ciphertext image.

Simulation experiments and theoretical analysis verify the effectiveness of this encryption algorithm against chosen plaintext attacks, violent attacks, statistical attacks, differential attacks, noise attacks, and cropping attacks. Therefore, the encryption algorithm proposed in this paper has good security performance.

## Figures and Tables

**Figure 1 entropy-25-01178-f001:**
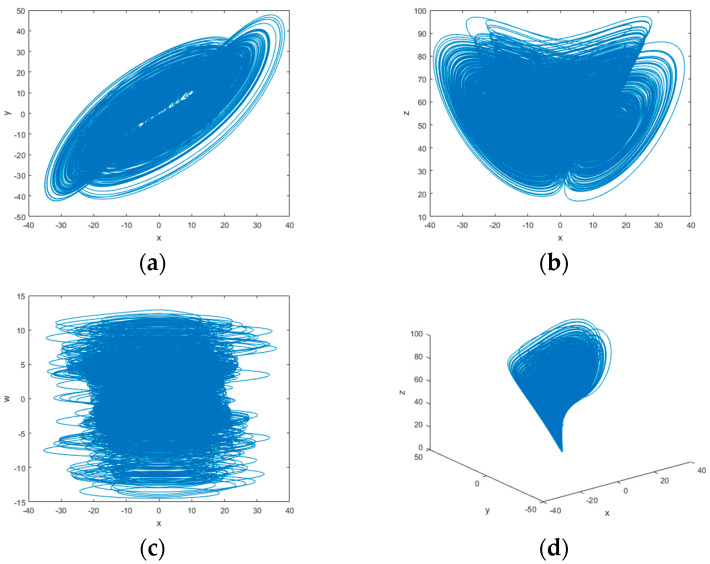
Phase diagram of the 4D hyperchaotic system: (**a**) *x*–*y* space; (**b**) *x*–*z* space; (**c**) *x*–*w* space; (**d**) *x*–*y–z* space.

**Figure 2 entropy-25-01178-f002:**
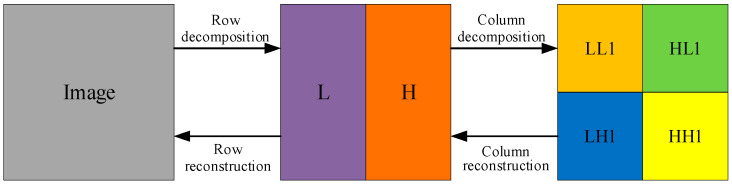
The DWT and IDWT process.

**Figure 3 entropy-25-01178-f003:**
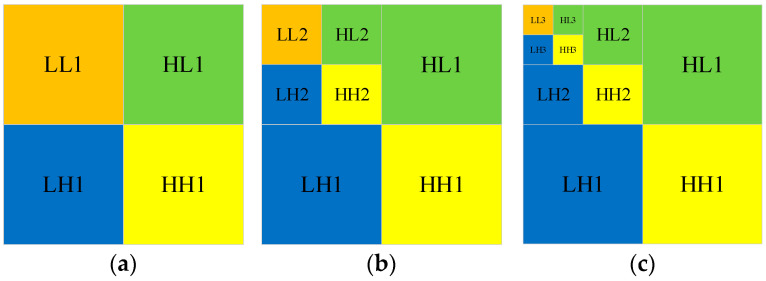
Schematic diagram of different levels of DWT: (**a**) One-level DWT; (**b**) Two-level DWT; (**c**) Three-level DWT.

**Figure 4 entropy-25-01178-f004:**
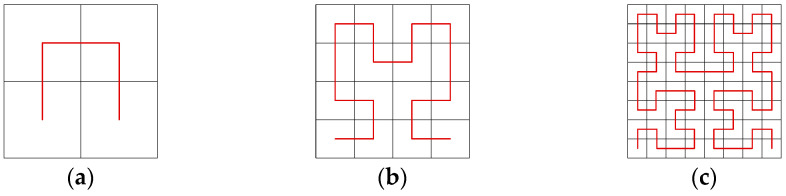
The Hilbert curve of different orders: (**a**) One-order Hilbert curve; (**b**) Two-order Hilbert curve; (**c**) Three-order Hilbert curve.

**Figure 5 entropy-25-01178-f005:**
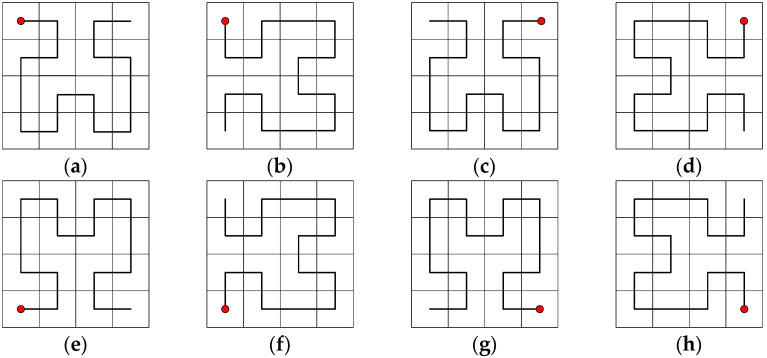
Schematic diagram of the eight modes of the Two-order Hilbert curve: (**a**) Mode1; (**b**) Mode2; (**c**) Mode3; (**d**) Mode4; (**e**) Mode5; (**f**) Mode6; (**g**) Mode7; (**h**) Mode8.

**Figure 6 entropy-25-01178-f006:**
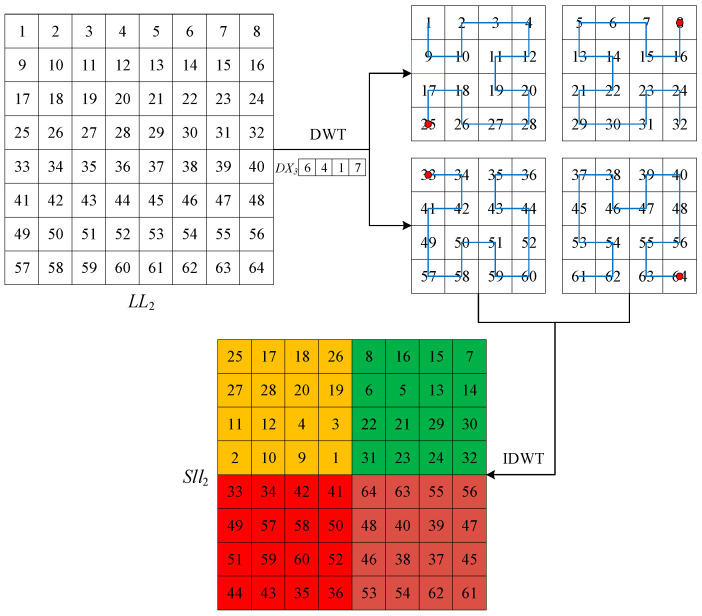
Schematic diagram of Hilbert curve scrambling process.

**Figure 7 entropy-25-01178-f007:**
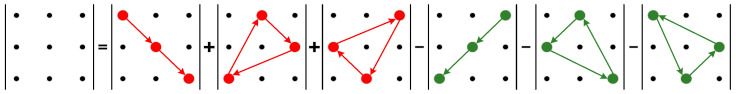
Third-order determinant expansion.

**Figure 8 entropy-25-01178-f008:**
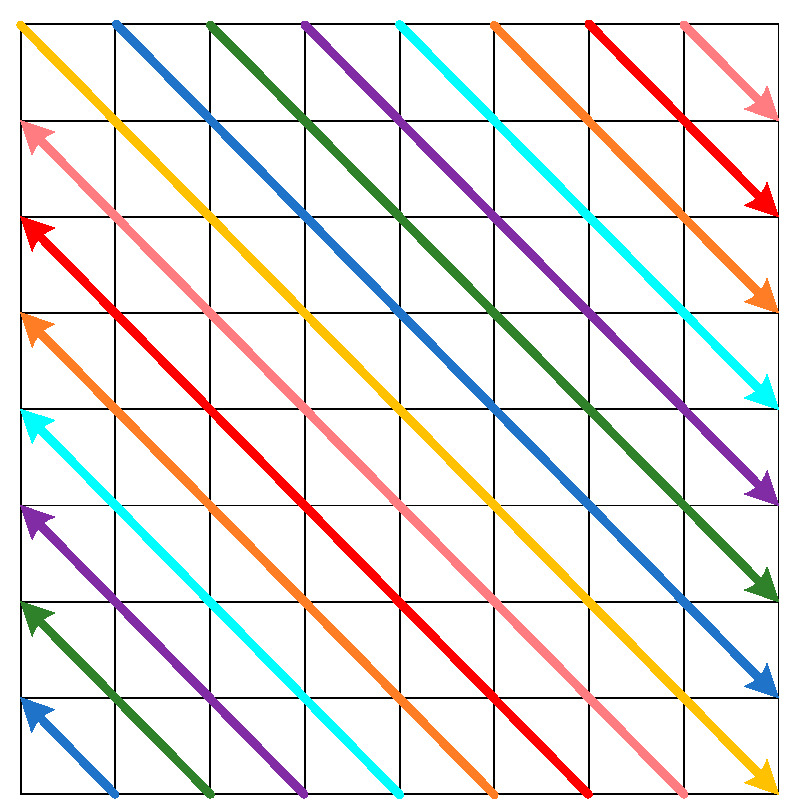
Main-diagonal extraction model.

**Figure 9 entropy-25-01178-f009:**
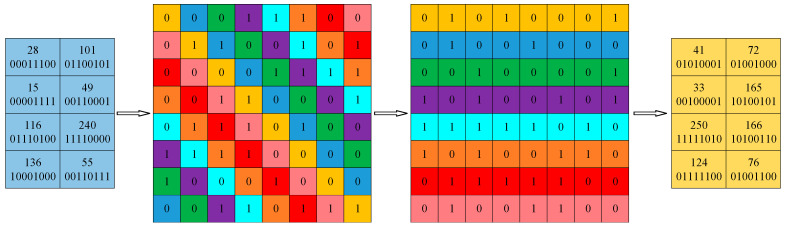
Example of the bit-level scrambling process.

**Figure 10 entropy-25-01178-f010:**
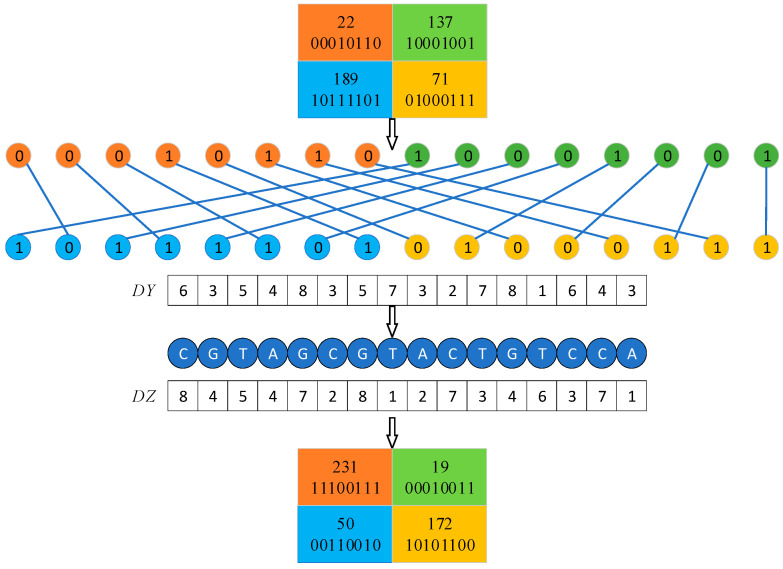
Dynamic DNA coding process.

**Figure 11 entropy-25-01178-f011:**
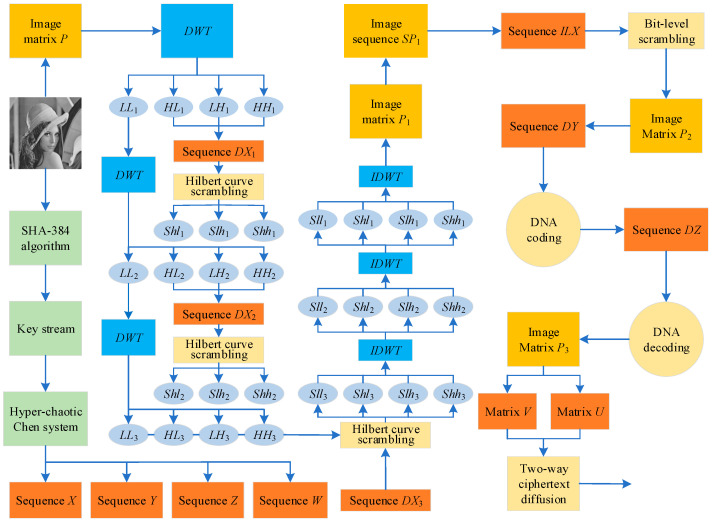
Encryption algorithm flow chart.

**Figure 12 entropy-25-01178-f012:**
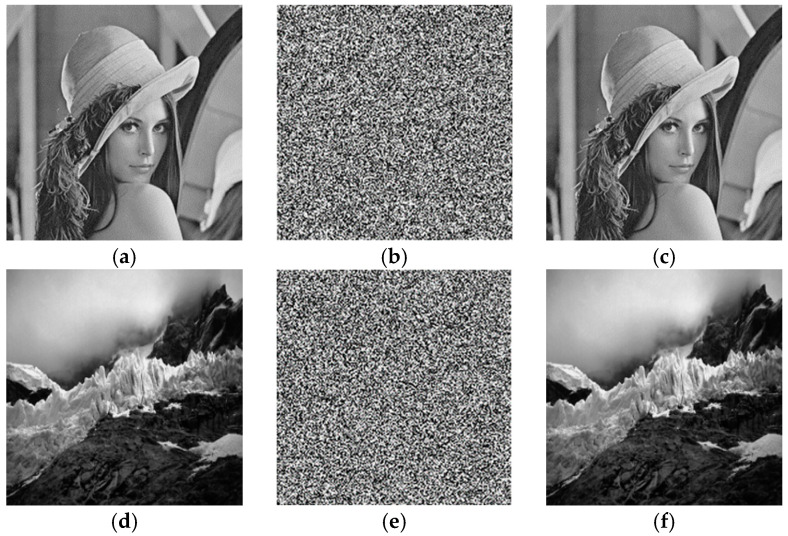
Simulation results: (**a**) Lena plaintext image; (**b**) Lena ciphertext image; (**c**) Lena decrypted image; (**d**) Jokul plaintext image; (**e**) Jokul ciphertext image; (**f**) Jokul decrypted image; (**g**) Bridge plaintext image; (**h**) Bridge ciphertext image; (**i**) Bridge decrypted image; (**j**) Bank plaintext image; (**k**) Bank ciphertext image; (**l**) Bank decrypted image; (**m**) Peppers plaintext image; (**n**) Peppers ciphertext image; (**o**) Peppers decrypted image.

**Figure 13 entropy-25-01178-f013:**
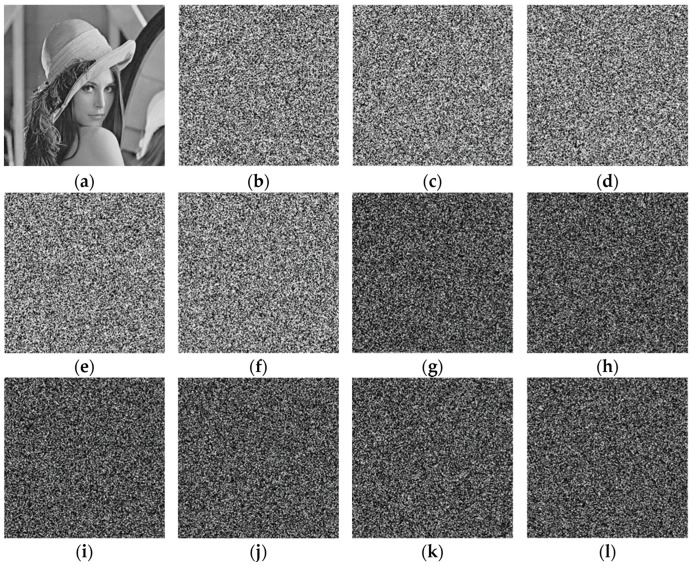
Key sensitivity test: (**a**) Lena; (**b**) Lena ciphertext; (**c**) $x_{0}+10^{-12}$; (**d**) $y_{0}+10^{-12}$; (**e**) $z_{0}+10^{-12}$; (**f**) $w_{0}+10^{-12}$; (**g**) Difference between (**c**,**d**); (**h**) Difference between (**c**,**e**); (**i**) Difference between (**c**,**f**); (**j**) Difference between (**d**,**e**); (**k**) Difference between (**d**,**f**); (**l**) Difference between (**e**,**f**).

**Figure 14 entropy-25-01178-f014:**
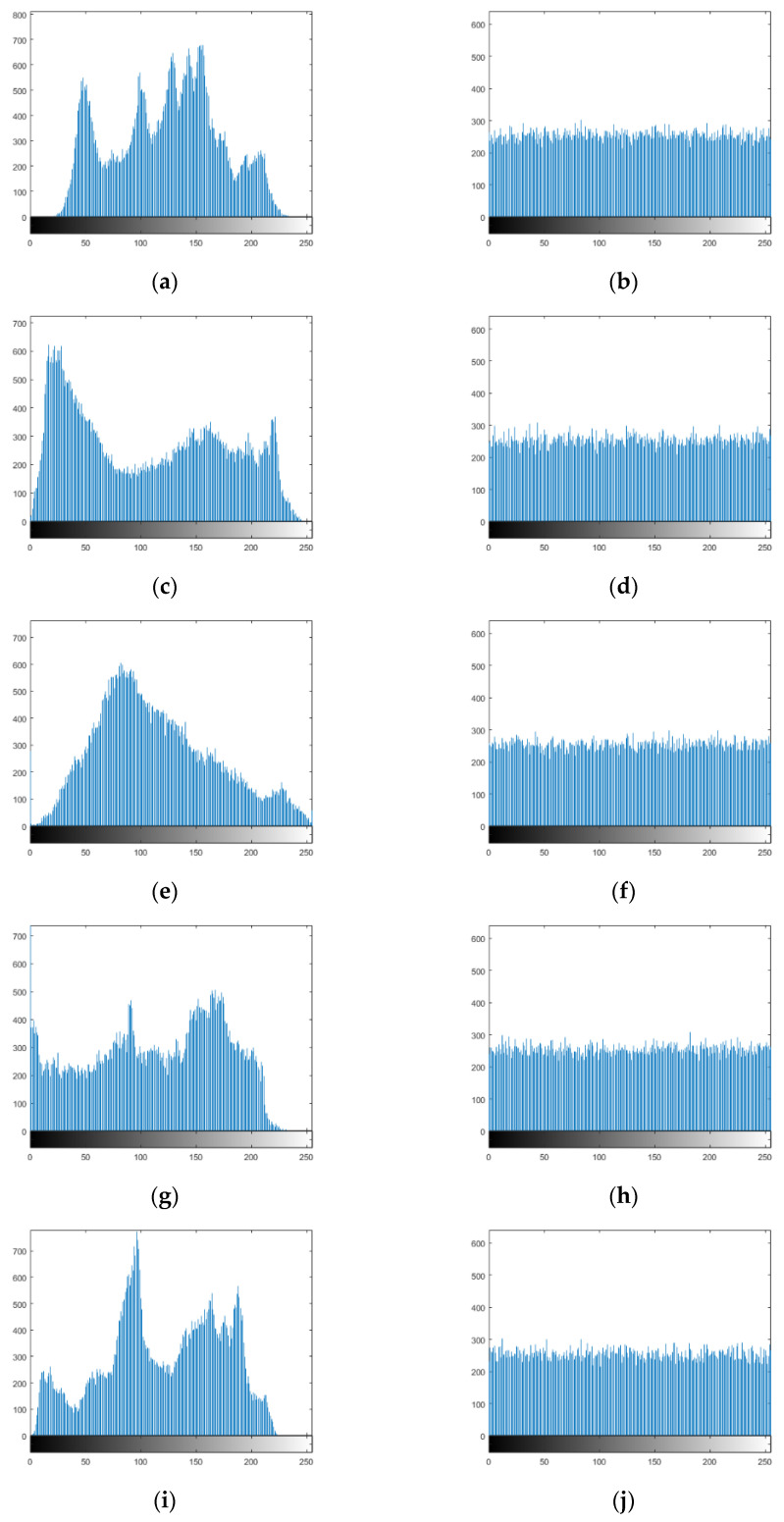
Histogram results: (**a**) Lena histogram; (**b**)Lena ciphertext histogram; (**c**) Jokul histogram; (**d**) Jokul ciphertext histogram; (**e**) Bridge histogram; (**f**) Bridge ciphertext histogram; (**g**) Bank histogram; (**h**) Bank ciphertext histogram; (**i**) Peppers histogram, (**j**) Peppers ciphertext histogram.

**Figure 15 entropy-25-01178-f015:**
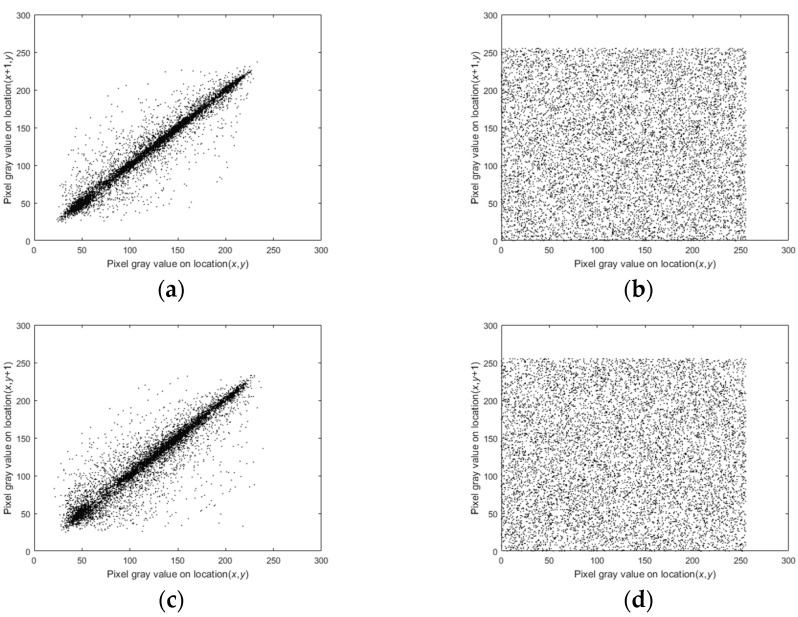
Correlation coefficients of Lena: (**a**) Horizontal correlation of the Lena plaintext image; (**b**) Horizontal correlation of the Lena ciphertext image; (**c**) Vertical correlation of the Lena plaintext image; (**d**) Vertical correlation of the Lena ciphertext image; (**e**) Diagonal correlation of the Lena plaintext image; (**f**) Diagonal correlation of the Lena ciphertext image.

**Figure 16 entropy-25-01178-f016:**
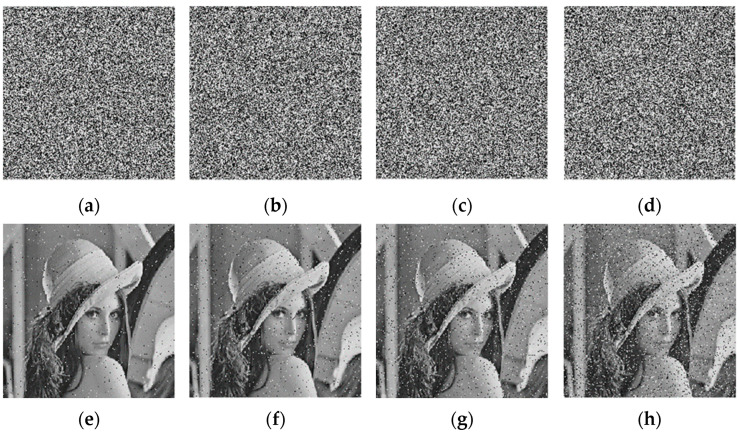
Noise attack experiment results: (**a**) 0.01 intensity ciphertext image; (**b**) 0.05 intensity ciphertext image; (**c**) 0.1 intensity ciphertext image; (**d**) 0.15 intensity ciphertext image; (**e**) 0.01 intensity decrypted image (**f**) 0.05 intensity decrypted image; (**g**) 0.1 intensity decrypted image; (**h**) 0.15 intensity decrypted image.

**Figure 17 entropy-25-01178-f017:**
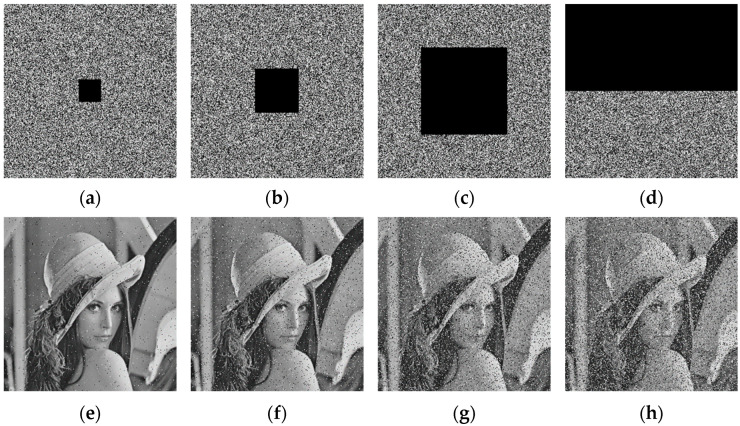
Cropping attacks experiment results: (**a**) 1/64 cropping; (**b**) 1/16 cropping; (**c**) 1/4 cropping; (**d**) 1/2 cropping; (**e**) Decrypted image of (**a**); (**f**) Decrypted image of (**b**); (**g**) Decrypted image of (**c**); (**h**) Decrypted image of (**d**).

**Table 1 entropy-25-01178-t001:** DNA coding rule.

Rule	1	2	3	4	5	6	7	8
00	A	A	T	T	C	C	G	G
01	G	C	G	C	T	A	T	A
10	C	G	C	G	A	T	A	T
11	T	T	A	A	G	G	C	C

**Table 2 entropy-25-01178-t002:** Chi-square test results.

Images	Lena	Jokul	Bridge	Bank	Peppers
plaintext	39,387	18,649	27,574	18,569	31,602
ciphertext	226.4821	234.8416	252.1406	267.2613	255.4587

**Table 3 entropy-25-01178-t003:** Correlation coefficients.

Images		Horizontal Direction	Vertical Direction	Diagonal Direction
Lena	plaintext	0.9663	0.9249	0.9194
	ciphertext	0.0013	0.0021	−0.0028
Jokul	plaintext	0.9777	0.9795	0.9635
	ciphertext	0.0024	−0.0035	−0.0029
Bridge	plaintext	0.9205	0.9402	0.8903
	ciphertext	−0.0016	0.0043	−0.0027
Bank	plaintext	0.9385	0.9235	0.8955
	ciphertext	0.0102	−0.0061	0.0054
Peppers	plaintext	0.9686	0.9651	0.9357
	ciphertext	0.0048	−0.0036	−0.0014

**Table 4 entropy-25-01178-t004:** Correlation coefficient comparison.

Images	Algorithms	Horizontal Direction	Vertical Direction	Diagonal Direction
Lena	Proposed	0.0013	0.0021	−0.0028
	[44]	−0.0059	−0.0046	−0.0003
	[14]	0.0060	0.0021	−0.005
	[45]	0.0015	−0.0090	−0.0120
	[46]	0.0058	−0.0051	−0.0030
Peppers	Proposed	0.0048	−0.0036	−0.0014
	[44]	−0.0026	−0.0012	−0.0050
	[14]	−0.0025	−0.0040	−0.0015
	[45]	−0.0083	0.0081	−0.0142
	[46]	−0.0011	−0.0073	−0.0019

**Table 5 entropy-25-01178-t005:** Information entropy.

Information Entropy	Lena	Jokul	Bridge	Bank	Peppers
Plaintext	7.4539	7.5209	7.4236	7.3841	7.5794
Ciphertext	7.9987	7.9985	7.9982	7.9977	7.9981

**Table 6 entropy-25-01178-t006:** Comparison of information entropy.

Images	Algorithms	Plaintext	Ciphertext
Lena	Proposed	7.4539	7.9987
	[44]	7.4492	7.9971
	[14]	7.4446	7.9974
	[45]	7.3875	7.9939
	[46]	7.4446	7.9974
Peppers	Proposed	7.5794	7.9981
	[44]	7.3576	7.9967
	[14]	7.3800	7.9972
	[45]	7.5697	7.9973
	[46]	7.5327	7.9967

**Table 7 entropy-25-01178-t007:** Local information entropy.

Images	Lena	Jokul	Bridge	Bank	Peppers
Local information entropy	7.9021	7.9023	7.9019	7.9031	7.9026
Pass or fail	Pass	Pass	Pass	Pass	Pass

**Table 8 entropy-25-01178-t008:** Homogeneity.

Images	Lena	Jokul	Bridge	Bank	Peppers
Plaintext	0.8456	0.8857	0.8237	0.8810	0.9002
Ciphertext	0.3895	0.3892	0.3877	0.3893	0.3907

**Table 9 entropy-25-01178-t009:** Comparison of homogeneity, contrast, and energy.

Algorithm	Homogeneity	Contrast	Energy
Proposed	0.3895	10.5331	0.0156
[49]	0.3896	10.4968	0.0156
[50]	0.3901	10.5324	0.0156
[51]	0.3887	10.5325	0.0156
[52]	0.4633	10.3011	0.0152

**Table 10 entropy-25-01178-t010:** Contrast.

Images	Lena	Jokul	Bridge	Bank	Peppers
Plaintext	0.5591	0.2997	0.4602	0.6230	0.2980
Ciphertext	10.5331	10.3813	10.3729	10.4099	10.4157

**Table 11 entropy-25-01178-t011:** Energy.

Images	Lena	Jokul	Bridge	Bank	Peppers
Plaintext	0.0786	0.0946	0.0838	0.0971	0.1135
Ciphertext	0.0156	0.0156	0.0156	0.0156	0.0156

**Table 12 entropy-25-01178-t012:** Comparison of MSE and PSNR values.

Algorithms	Images	MSE	PSNR
Proposed	Jokul	8395.9526	8.8901
	Bridge	7730.2871	9.2488
	Bank	9505.5392	8.3510
	Peppers	8312.6971	8.9334
	Lena	10,659.0009	7.8536
[45]	Lena	7752.6	9.2363
[53]	Lena	9056.1634	8.5613
[54]	Lena	7802.1	9.2087
[55]	Lena	7797.7	9.2111

**Table 13 entropy-25-01178-t013:** NPCR and UACI values.

Size	Images	NPCR (%)	UACI (%)
256×256	Lena	99.6073	33.4682
	Jokul	99.6154	33.4557
	Bridge	99.5886	33.4564
	Bank	99.6012	33.4623
	Peppers	99.5979	33.4708
512×512	Lena	99.6023	33.4658
	Jokul	99.6135	33.4717
	Bridge	99.5963	33.4576
	Bank	99.6155	33.4698
	Peppers	99.6004	33.4618

**Table 14 entropy-25-01178-t014:** Comparison of NPCR and UACI values.

Size	Algorithm	NPCR (%)	UACI (%)
Lena (256×256)	Proposed	99.6073	33.4682
	[44]	99.6197	33.0443
	[14]	90.6047	33.4719
	[45]	99.6185	33.4561
	[46]	99.6085	33.4633
Lena (512×512)	Proposed	99.6023	33.4618
	[45]	99.6044	33.4117
	[56]	99.6101	33.4945
	[57]	99.6002	33.5079
	[58]	99.6140	31.4646

## Data Availability

Not applicable.

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
