# Peer review of "An Image Encryption Algorithm Based on Improved Hilbert Curve Scrambling and Dynamic DNA Coding"

_entropy, 2023, doi:10.3390/e25081178_

Round 1
Reviewer 1 Report
There is a typo in the title: it should be "Scrambling". Also, please check the entire paper for typos: see e.g. Line 128 (a missing space between "where" and delta).
I do not see the need of writing "Ref." before citing a reference, this is not the usual expression.
Please ensure that you have analyzed the proposed scheme from the security perspective as many of the current image encryption algorithms are weak and can be quickly broken.
Statistical methods are not merely enough in this case. You are not dealing with steganography, but encryption.
Please either use suitable cryptanalysis methods to analyze your scheme or prove the security.
Please check the paper for typos, I have spotted a few.
Reviewer 2 Report
Abstract: An algorithm using Hilbert curve and DNA coding for image encryption was proposed, and inverse discrete wavelet transform (IDWT) technology was employed to overcome the limitations of conventional permutation security techniques. The experimental results showed that the image encryption algorithm effectively resists statistical attacks and differential attacks, which confirms good security and robustness.
The advantages of this algorithm are as follows:
1. High level of security: the Hilbert curve-based algorithm effectively resists statistical and differential attacks, since no significant changes are made to the already encrypted data.
2. Efficient data processing: The processing speed for large amounts of data is fast, as the pixel value processing is based on the Hilbert curve and DNA coding.
3. Permutation and diffusion: the encryption algorithm performs permutation and diffusion in the spatial and frequency domains, overcoming the limitations of conventional permutation.
The disadvantages of this algorithm are as follows. Is there any way to improve these drawbacks?
1. The input image must be a square. If a non-square image is input to which this algorithm cannot be applied, additional processing is required.
2. The processing speed of the system may slow down due to the computational load generated by the encryption algorithm. When this algorithm is used to process a large amount of data, the disadvantage is that the computation speed is slower.
3. The algorithm based on the Hilbert curve is subject to the constraint that the data to be processed must be a power of 2 and the sizes of width and height should be equal. Thus, in order to apply this algorithm, the input image must satisfy these conditions.
